# Unlearning is Better than Unseen: Unlearning Score-based Generative Model

## Abstract

Diffusion generative models, including Score-Based Generative Models (SGM) and Denoising Diffusion Probabilistic Models (DDPM), have demonstrated remarkable performance across various domains in recent years. However, concerns regarding privacy and potential misuse of AI-generated content have become increasingly prominent. While generative unlearning methods have been investigated on DDPM models, research on unlearning SGM is still largely missing. Furthermore, the current 'gold standard' of machine unlearning—retraining a model from scratch after removing the undesirable data, does not perform well in SGM and its downstream tasks, such as image inpainting and reconstruction. To fill this gap, we propose the first Score-based Generative Unlearning (SGU) for SGM, which surpasses the previous 'gold standard' of unlearning. SGU introduces a new score adjustment strategy that deviates the learned score from the original undesirable data score during the continuous-time stochastic differential equation process. Extensive experimental results demonstrate that SGU significantly reduces the likelihood of generating undesirable content while preserving high quality for normal image generation. Albeit designed for SGM, SGU is a general and flexible unlearning framework that is compatible with diverse diffusion architectures (SGM and DDPM) and training strategies (re-training and fine-tuning), and enables zero-shot transfer of the unlearning generative model to downstream tasks, including image inpainting and reconstruction. The code will be shared upon acceptance.

## 1 Introduction

The development and application of generative artificial intelligence technology have sparked a new wave of interest in AI. Recent advancements in deep generative models have made it possible to generate highly realistic images. Two types of diffusion generative models, namely Score-Based Generative Models (SGM) (Song et al., 2021) and Denoising Diffusion Probabilistic Models (DDPM) (Ho et al., 2020), represent the state-of-the-art methods in this field. These models sequentially perturb the training data with gradually increasing noise and then learn to reverse this perturbation. They effectively address several challenges that previous generative techniques faced, such as aligning the posterior distribution in Variational Autoencoders (VAEs)(Kingma & Welling, 2013; Wang et al., 2021), handling the instability of adversarial objectives in Generative Adversarial Networks (GANs)(Goodfellow et al., 2020; Wang et al., 2022), reducing the high training time and computational costs of Markov Chain Monte automobilelo (MCMC) methods in Energy-Based Models (EBMs)(LeCun et al., 2006; Gao et al., 2020), and alleviating network constraints in normalizing flows(Dinh et al., 2016; Zhang & Chen, 2021).

However, despite these technological breakthroughs, diffusion generative models also pose risks of privacy breaches and potential misuse, raising public concerns regarding privacy, copyright and the dissemination of misinformation (Dubiński et al., 2024). Firstly, diffusion generative models possess memorization capabilities (Somepalli et al., 2023), which can lead to the replication of all or part of the training data, resulting in privacy breaches within the training dataset. In addition to privacy concerns, diffusion generative models are also susceptible to misuse, potentially producing inappropriate digital content, such as deepfakes that can be used to create misleading videos or images of individuals, potentially damaging their reputation or spreading misinformation (Rando et al., 2022; Salman et al., 2023; Schramowski et al., 2023). Additionally, diffusion generative models can imitate various artistic styles(Shan et al., 2023; Gandikota et al., 2023). Unauthorized use of others' portraits

or artworks for synthesis may infringe upon portrait and intellectual property rights, raising legal concerns. These issues can negatively impact mental health, blur the line between reality and fiction, and potentially erode social trust and values. Therefore, ensuring that AI technology advances human and societal development without causing harm is a critical and urgent challenge.

To address this challenge, Machine Unlearning (MU) mechanism can enable generative models to forget data deemed Not Suitable For Generation (NSFG), thereby protecting copyright and preventing the generation of harmful content (Schramowski et al., 2023; Wang et al., 2023; Li et al., 2024; Shaik et al., 2023). Very recently, MU for conditional DDPM has started to appear on text-conditioned image generation (Gandikota et al., 2023; Heng & Soh, 2023; Fan et al., 2024; Zhang et al., 2024; Kumari et al., 2023; Wu et al., 2024; Heng & Soh, 2024), but unlearning in score-based generative model has been largely absent so far. Score-based generative models offer notable benefits in stability, sampling efficiency, and generation quality, making them increasingly favored in practical use cases. Therefore, this paper aims to fill this research gap by proposing a new unlearning score-based generative model.

Unlearning for score-based generative model presents several challenges. First, the state-of-the-art MU method is to re-train a model from scratch after removing the undesirable data from the original training data, a process we refer to as *Unseen Re-training*. This method is often regarded as the 'gold standard' in MU. However, we have observed that even after *Unseen Re-training*, score-based generative models can still produce undesirable content(Figure 1). Moreover, the current 'gold standard' does not work well in downstream tasks, such as image inpainting and reconstruction. How to design a better unlearning method that can exceed the current MU 'gold standard' in score-based generative model is an crucial challenge. Furthermore, most existing unlearning generative research focuses on conditional generative models (Gandikota et al., 2023; Heng & Soh, 2023; Fan et al., 2024; Zhang et al., 2024; Kumari et al., 2023; Wu et al., 2024; Heng & Soh, 2024), especially on text-to-image generation with conditional DDPM. These unlearning methods rely heavily on specific conditions, such as text prompts, which limits the generalizability of their unlearning frameworks. These unlearning methods are tightly coupled with the specific condition (e.g., text prompts) (Gandikota et al., 2023; Heng & Soh, 2023), which limits the generalizability of their unlearning frameworks. In contrast, unconditional models form the basis for many generative frameworks, including conditional generators that are often built upon unconditional architectures. Developing unlearning strategies for unconditional models could therefore provide more general solutions that apply to a broader range of generative models. While unlearning on unconditional VAEs and GANs have been investigated very recently (Moon et al., 2024), unlearning on unconditional SDG is still under-explored. Finally, unlike most unlearning methods for DDPMs, which aim to reduce the evidence lower bound (ELBO) on the distribution of the forgotten data, score-based generative models focus on estimating the score of the data distribution across a continuous noise schedule. How to design an effective unlearning method from a score-based perspective remains unexplored.

To address the challenges, we propose the first Score-based Generative Unlearning (SGU) method for SGM, which surpasses the previous 'gold standard' for generative unlearning in SGM. SGU aims to overcome the limitations of current 'gold standard' in score-based generative unlearning, by introducing a straightforward yet effective strategy to alter the score function. Our key idea is to deviate the learned score from the original NSFG data score during the continuous SDE process, while ensuring that it approximates the SFG data score to maintain generation quality. We present two variants of SGU to handle different unlearning scenarios. Since the score is defined as the gradient of the logarithm of the probability density function, the first variant is to learn a score $s_\theta^{\mathbf{u}}(\mathbf{x}, \mathrm{t})$ that is orthogonal to the $\nabla_{\mathbf{x}^f} \log p_t(\mathbf{x}^f)$ of the ground truth NSFG data distribution. This orthogonality ensure $s_\theta^{\mathbf{u}}(\mathbf{x}, \mathrm{t})$ and $\nabla_{\mathbf{x}^f} \log p_t(\mathbf{x}^f)$ are uncorrelated, helping to prevent the generation of undesired content. Next, in cases where the NSFG and SFG distributions are very similar, or when content needs to be erased from a pre-trained model, we propose another variant of USGM in which the learned score $s_\theta^{\mathbf{u}}(\mathbf{x}, \mathrm{t})$ is the inverse of $\nabla_{\mathbf{x}^f} \log p_t(\mathbf{x}^f)$. This inverse relationship helps to more effectively negate the influence of the NSFG data, improving convergence during training. Albeit designed for SGM, SGU is a general and flexible unlearning framework that is compatible with diverse diffusion architectures (SGM and DDPM) and training strategies (re-training and fine-tuning), and enables zero-shot transfer of the unlearning generative model to downstream tasks, including image inpainting and reconstruction. Extensive results demonstrate that SGU performs well in category forgetting, feature forgetting, and various downstream tasks such as image inpainting and reconstruction.

## 2 PRELIMINARIES

### 2.1 GENERATIVE MODELING

A generative model is often a statistical model $p_\theta(\mathbf{x})$, where $\theta \in \Theta$, with $\theta$ representing the model parameters and $\Theta$ denoting the set of allowable parameter values. The goal of a generative model is to learn and estimate the unknown data distribution $p_{data}(\mathbf{x})$ from a given dataset, allowing us to generate new data samples and query the probability of any data point ideally. We find the optimal parameter $\theta^* \in \Theta$ such that $p_\theta(\mathbf{x}) \approx p_{data}(\mathbf{x})$. When the statistical model $p_{\theta*}(\mathbf{x})$ closely approximates the data distribution $p_{\text{data}}(\mathbf{x})$, we can use $p_{\theta*}(\mathbf{x})$ as a proxy for $p_{\text{data}}(\mathbf{x})$ to generate new samples and evaluate probability values. To this end, we can use a distance measure such as KL divergence to quantify the difference between two distributions, $p_\theta(\mathbf{x})$ and $p_{\text{data}}(\mathbf{x})$. This allows us to determine the optimal parameter $\theta^*$, which can be simply formulated as follows:

$$\theta^* = \arg\min_{\theta \in \Theta} \text{KL}(p_{\text{data}}(\mathbf{x})||p_\theta(\mathbf{x})) = \mathbb{E}_{p_{\text{data}}(\mathbf{x})}\left[\frac{p_{\text{data}}(\mathbf{x})}{p_\theta(\mathbf{x})}\right], \tag{1}$$

wherein the expectation can be estimated using the empirical mean over samples in the training dataset. Therefore, we can train a generative model $p_\theta(\mathbf{x})$ on the dataset consisting of independent and identically distributed (i.i.d.) samples $\{\mathbf{x}_i \in \mathbb{R}^D\}_i^N$ from $p_{data}(\mathbf{x})$ by maximizing the average log-likelihood across the training data points, which is known as maximum likelihood estimation (MLE), i.e.,

$$\theta^* = \arg\max_{\theta \in \Theta} \mathbb{E}_{p_{\text{data}}(\mathbf{x})} \log p_\theta(\mathbf{x}). \tag{2}$$

**Machine unlearning in Generative Model** Let $\mathcal{D} = \{\mathbf{x}_i\}_i^N \in \mathbb{R}^D$ be the training data, which follow the distribution $\mathbf{x}_i \sim p_d$. Let $\mathcal{D}_f = \{\mathbf{x}_i^u\}_i^M \subseteq D$ denote the forgetting dataset containing privacy or toxicity issues, which is referred to as not suitable for generation (NSFG) data, following the distribution $p_f(\mathbf{x})$. The remaining data, $\mathcal{D}_g = \mathcal{D}\backslash\mathcal{D}_f = \{\mathbf{x}_i^g\}_i^{N-M} \sim p_g(\mathbf{x})$, represents the Suitable For Generation (SFG) data. Our goal is to enable the generative model to avoid generating NSFG samples while maintaining the quality of image generation for SFG data. We refer to such a generative model as an unlearning generative model. We use the symbol $p$ to denote either a probability distribution or its probability density or mass function depending on the context.

### 2.2 SCORE-BASED GENERATIVE MODELING WITH SDES

**Score model** *Score function* is the abbreviation of *Stein score function* (Stein, 1972). It is defined as the gradient of the log density of a probability distribution. Specifically, the corresponding score function $s(\mathbf{x})$ for the probability density function is given by $\nabla_\mathbf{x} \log p_t(\mathbf{x})$. Given a probability density function, its score function is uniquely determined by the gradient of the log-density. Conversely, a given score function can be used to recover the corresponding density function. Thus, the score function retains the same amount of information as the probability density function. We refer to a model that represents a score function as a score model, denoted by $s_\theta(\mathbf{x})$, where $\theta$ represents the model parameters. The score function does not require calculating the normalization constant, which is a major advantage over the density function. As a result, it is considerably easier to model using flexible deep neural networks.

**Score-Based Generative Modeling with SDEs** The two main components of a score-based SDE generative model are the *forward process* and the *reverse process*.

The forward process $\{\mathbf{x}(t) \in \mathbb{R}^d\}_{t=0}^T$ transforming data from the distribution $p_{data}(\mathbf{x})$ to a simple noise distribution with a continuous-time stochastic differential equation (SDE)

$$d\mathbf{x} = \mathbf{f}(\mathbf{x}, t)dt + g(t)d\mathbf{w}, t \in [0, T], \tag{3}$$

where $\mathbf{f} : \mathbb{R}^d \to \mathbb{R}^d$ is called the drift coefficient of the SDE, $g \in \mathbb{R}$ is called the diffusion coefficient, and $\mathbf{w}$ represents the standard Brownian Motion. Let $p_t(\mathbf{x})$ denote the density of $\mathbf{x}(t)$. At time $t = 0$, the initial distribution of $\mathbf{x}(0)$ follows $p_0 := p_{data}$, while at time $t = T$, $\mathbf{x}(T)$ adheres to $p_T$. Here, $p_T$ commonly represents a prior distribution known for its manageable form and ease of sampling, frequently taking the shape of a Gaussian distribution.

The reverse process then converts noise into samples via reversing the diffusion process, effectively executing generative modeling. Remarkably, $\mathbf{x}(t)$ satisfies a reverse-time SDE:

$$dx = [\mathbf{f}(\mathbf{x}, t) - g^2(t)\nabla_\mathbf{x} \log p_t(\mathbf{x})]dt + g(t)d\bar{\mathbf{w}}, \tag{4}$$

where $\bar{\mathbf{w}}$ is a Brownian motion in the reverse time direction, and $dt$ represents an infinitesimal negative time step.

Running the reverse process requires estimating the score function of the law of the forward process. this is typically done by training neural networks on a score-matching objective.

**Score Estimation**    In practice, when we only have sample access to $p_{data}$, the score function $\nabla_\mathbf{x} \log p_t(\mathbf{x})$ is not available. We can train a time-dependent score-based model $s_\theta(\mathbf{x}, \mathrm{t})$, to approximate $\nabla_\mathbf{x} \log p_t(\mathbf{x})$, using the following weighted sum of denoising score matching objectives

$$\min_\theta \mathbb{E}_t \, L_{SGM} = \min_\theta \mathbb{E}_t \lambda(t) \mathbb{E}_{\mathbf{x}(0)} \mathbb{E}_{\mathbf{x}(t)} \|s_\theta(\mathbf{x}(t), t) - \nabla_{\mathbf{x}(t)} \log p_{0t}(\mathbf{x}(t) \mid \mathbf{x}(0))\|_2^2 \tag{5}$$

where $\mathbf{x}(0) \sim p_0(\mathbf{x})$ and $\mathbf{x}(t) \sim p_{0t}(\mathbf{x}(t) \mid \mathbf{x}(0))$, $t \sim \mathcal{U}(0, T)$ is a uniform distribution over $[0, T]$, $p_{0t}(\mathbf{x}(t) \mid \mathbf{x}(0))$ denotes the transition probability from $\mathbf{x}(0)$ to $\mathbf{x}(t)$, and $\lambda(t) \in \mathbb{R}_{>0}$ denotes a positive weighting function. Note that  Equation (5) uses denoising score matching, but other score matching objectives, such as sliced score matching (Song et al., 2020) and finite-difference score matching (Pang et al., 2020) are also applicable here.

## 3    Unlearning is Better than Unseen

### 3.1    Motivation

Two mainstream strategies for generative unlearning involve either erasing learned NSFG content from a pre-trained generator, *i.e.*, *Erased Fine-tuning*, or re-training the generator from scratch after removing the forgetting data $D_g$ from original training dataset, *i.e.*, *Unseen Re-training*. *Erased Fine-tuning* modifies specific parts of a generative model (*e.g.*, weights or learned features) to forget the influence of specific content, yet may still inadvertently generate unwanted contents (Qi et al., 2023). From a data representation and gradient space perspective, research has demonstrated that when fine-tuning data contains examples closely resembling known explicit content in any feature space, the model's susceptibility to adversarial attacks increases, resulting in the generation of NSFG content (He et al., 2024). Conversely, *Unseen Re-training* is regarded as the state-of-the-art generative unlearning strategy, significantly outperforming *Erased Fine-tuning* (Xu et al., 2024). However, contrary to the belief that *Unseen Re-training* is considered the 'gold standard' for data forgetting (Thudi et al., 2022; Fan et al., 2024), we found that when the distribution distance between $D_g$ and $D_f$ is close, malicious users may still exploit the model to generated undesirable content. Similarly, Gandikota et al. (2023) demonstrates that even re-training the SD 2.0 (Rombach et al., 2022b) model on filtered datasets that exclude explicit images, explicit content persists in the model's outputs using prompts from the Inappropriate Image Prompts (I2P) benchmark (Schramowski et al., 2023).

Figure 1 shows a toy example to illustrate the above phenomenon. We train a Variance Exploding Stochastic Differential Equation (VE SDE) model (Song et al., 2021) on the dataset $D$ sampled from a mixture of two-dimensional Gaussian distributions, where the data distribution (shown on the left of Figure 1 (a) & (c) is defined as $p_{\mathrm{data}} = \frac{4}{5}\mathcal{N}((-2,-2), I) + \frac{2}{5}\mathcal{N}((0,0), I) + \frac{4}{5}\mathcal{N}((2,2), I)$. We refer to this standard trained VE SDE as *Standard VE SDE*, and it learns a data distribu-

Table 1: The Negative log-likelihood (NLL) values of different methods with respect to the data from $p_{\mathrm{data}}$.

| Test | Standard | Unseen | Unlearning |
|------|----------|--------|------------|
| $\mathcal{D}_g$ | 10.91 | 10.63 | 10.64 |
| $\mathcal{D}_f$ | 10.73 | 11.59 | 39.01 |

tion as is shown in Figure 1 (a) on the right. NSFG data $D_f$ is identified as the data from distribution $p_{\mathrm{f}} = \mathcal{N}((0,0), I)$ (colored green in Figure 1 (a) on the left). The remaining data sampled from $p_{\mathrm{g}} = \frac{4}{5}\mathcal{N}((-2,-2), I) + \frac{4}{5}\mathcal{N}((2,2), I)$ represent the SFG data (colored red in Figure 1 (a) on the left), denoted as $D_g$. The generator trained only on $D_g$ is referred to as *Unseen Re-training*. Its learned data distribution is shown in Figure 1 (b) on the right. Figure 1 (b) demonstrates that NSFG data were inadvertently generated even when the model was trained on pure SFG data.

Additionally, we quantify the generation probability of NSFG and SFG data in terms of Negative Log-likelihood (NLL) given different generators in Table 1. For *Standard VE SDE*, it is reasonable

for $D_g$ and $D_f$ to have the same likelihood, as both data are observed during training. However, for *Unseen Re-training*, the likelihood of $D_f$ is almost the same as $D_g$. This indicates that the generator trained by *Unseen Re-training* did not use NSFG data for training, it can still fit the unseen data well. This raises our concerns about the "gold standard" of machine unlearning in generative models, as the unlearning model can still generate undesirable content even if it has never seen $D_f$. Naturally, a question occurs to us:

***Is there a better unlearning method that can exceed the current 'gold standard' for generative unlearning, enabling the generator to entirely forget undesirable contents rather than merely 'unseen' them?***

### 3.2 SCORE-BASED GENERATIVE UNLEARNING

To answer the above question, we first formalize the *Unseen Re-training* process as follows:

$$\theta^* = \arg\min_{\theta \in \Theta} \left( D_{\mathrm{KL}}(p_g(\mathbf{x}) \| p_\theta(\mathbf{x})) \right). \tag{6}$$

An *Unseen Retrained* generator only approximates $p_g(\mathbf{x})$ and the model generates data that follows $p_g(\mathbf{x})$ with high likelihood as described in Equation (6). However, Equation (6) does not consider the likelihood of generating $D_f$. If the distributions $p_f(\mathbf{x})$ and $p_g(\mathbf{x})$ are close or overlapping, *Unseen Re-training* may not control the probability of generating $D_f$ (see Table 1). Therefore, we propose a new *Unlearning Re-training* strategy to prevent the generator from generating undesired content by maximizing the distance between $p_f(\mathbf{x})$ and $p_\theta(\mathbf{x})$, while minimizing the distance between $p_g(\mathbf{x})$ and $p_\theta(\mathbf{x})$, *i.e.*,

$$\theta^* = \arg\min_{\theta \in \Theta} \left\{ D_{\mathrm{KL}}(p_g(\mathbf{x}) \| p_\theta(\mathbf{x})) - D_{\mathrm{KL}}(p_f(\mathbf{x}) \| p_\theta(\mathbf{x})) \right\}. \tag{7}$$

Different from *Unseen Retrained* model only approximating $p_g(\mathbf{x})$, the objective of *unlearning Re-training* is to force the unlearning model to assign low likelihood to $p_f(\mathbf{x})$ and high likelihood to $p_g(\mathbf{x})$. Considering that machine unlearning for score-based generative model has not been investigated, we focus on unlearning in score-based generative model and instantiate *unlearning Re-training* from a perspective of score estimation. It is well known that score estimation plays a crucial role in the generation process of score-based generative models. Theoretically, as long as the score estimation is sufficiently accurate and the forward diffusion time is long enough (such that the final noise distribution approaches the prior distribution), diffusion models can approximate any continuous data distribution with polynomial complexity under weak conditions (Chen et al., 2023a). Therefore, Equation (7) can be framed as a score estimation problem, where different score functions are estimated for $p_g(\mathbf{x})$ and $p_f(\mathbf{x})$. The question now becomes how to train a time-dependent score-based model $s_\theta^{\mathbf{u}}(\mathbf{x}, \mathrm{t})$ to approximate $\nabla_{\mathbf{x}^g} \log p_t(\mathbf{x}^g)$ and deviation $\nabla_{\mathbf{x}^f} \log p_t(\mathbf{x}^f)$. For approximating $p_g(\mathbf{x})$, we can directly use the original score estimation:

$$L_{USGM(g)} = \lambda(t) \mathbb{E}_{\mathbf{x}(0)} \mathbb{E}_{\mathbf{x}(t)} [\| s_\theta^{\mathbf{u}}(\mathbf{x}^g(t), t) - \nabla_{\mathbf{x}^g(t)} \log p_{0t}(\mathbf{x}^g(t) \mid \mathbf{x}^g(0)) \|_2^2], \ \mathbf{x}^g \in \mathcal{D}_g. \tag{8}$$

For unlearning $p_f(\mathbf{x})$, if the estimated score at any moment deviates from the score of the NSFG data on the timeline from 0 to $T$, the samples generated during sampling will be far away from the data distribution of NSFG. Under this goal, a straightforward idea is to reduce the correlation between $s_\theta^{\mathbf{u}}(\mathbf{x}, \mathrm{t})$ and $s_\theta^{\mathbf{u}}(\mathbf{x}, \mathrm{t})$, i.e. minimizing the inner product of the two scores:

$$L_{USGM(f)} = \lambda(t) \mathbb{E}_{\mathbf{x}(0)} \mathbb{E}_{\mathbf{x}(t)} [\| s_\theta^{\mathbf{u}}(\mathbf{x}^f(t), t) \cdot \nabla_{\mathbf{x}^f(t)} \log p_{0t}(\mathbf{x}^f(t) \mid \mathbf{x}^f(0)) \|_2^2], \ \mathbf{x}^f \in \mathcal{D}_f. \tag{9}$$

Equation (9) seeks for the null space of $\nabla_{\mathbf{x}^f} \log p_t(\mathbf{x}^f)$, so that for $\mathbf{x}^f \in \mathcal{D}_f$, $s_\theta^{\mathbf{u}}(\mathbf{x}, \mathrm{t}) \cdot \nabla_{\mathbf{x}^f} \log p_t(\mathbf{x}^f) \to 0$. We refer to this unlearning optimization as *Orthogonal Unlearning*. However, in our preliminary experiments, we observed that when $p_g(\mathbf{x})$ and $p_f(\mathbf{x})$ are very close (e.g. when generating human faces where local features like bangs or beards are undesirable) or when $s_\theta^{\mathbf{u}}(\mathbf{x}^{\mathrm{f}}, \mathrm{t})$ has been learned well (e.g. erasing undesirable content from a converged pre-trained generator), limiting the search to the null space of $\nabla_{\mathbf{x}^f} \log p_t(\mathbf{x}^f)$ becomes difficult to optimize. To address this issue, we expand the search space by ensuring $s_\theta^{\mathbf{u}}(\mathbf{x}^f(t), t) \cdot \nabla_{\mathbf{x}^f} \log p_t(\mathbf{x}^f) < 0$, $\mathbf{x}^f \in \mathcal{D}_f$. This leads us to define a new unlearning objective called *Obtuse Unlearning*:

$$L_{USGM(f)} = \lambda(t) \mathbb{E}_{\mathbf{x}(0)} \mathbb{E}_{\mathbf{x}(t)} [s_\theta^{\mathbf{u}}(\mathbf{x}^f(t), t) \cdot \nabla_{\mathbf{x}^f(t)} \log p_{0t}(\mathbf{x}^f(t) \mid \mathbf{x}^f(0))], \ \mathbf{x}^f \in \mathcal{D}_f. \tag{10}$$

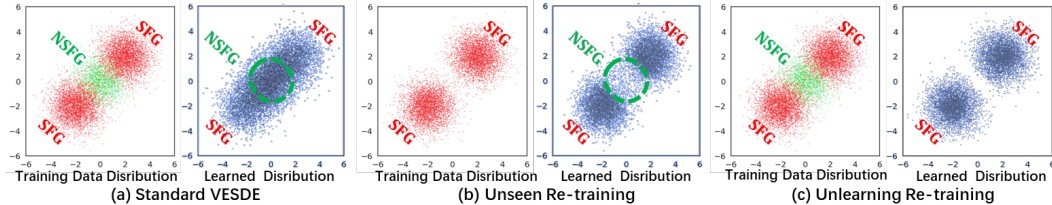

Figure 1: The samples from the mixture Gaussian distribution and the samples generated by the model trained by *Standard VE SDE* (a), *Unseen Re-training* (b) and *Unlearning Re-training* (c). The left side of (a), (b) and (c) represents the training data, in which the green part is NSFG data, and the red part is SFG data. The right side of (a), (b) and (c) represents the data generated by diffusion models.

The final loss of unlearning score-based generative modeling can be expressed as:

$$\min_{\theta} \mathbb{E}_{t \sim \mathcal{U}(0,T)} L_{USGM} = \min_{\theta} \mathbb{E}_{t \sim \mathcal{U}(0,T)} \left( \alpha L_{USGM(g)} + (1 - \alpha) L_{USGM(f)} \right), \quad (11)$$

where $\mathcal{U}(0,T)$ is a uniform distribution over $[0,T]$, $p_{0t}(\mathbf{x}(t) \mid \mathbf{x}(0))$ denotes the transition probability from $\mathbf{x}(0)$ to $\mathbf{x}(t)$, $\lambda(t) \in \mathbb{R}_{>0}$ denotes a positive weighting function and $\alpha$ is a hyperparameter whose value depends on the ratio of $M$ to $N$.

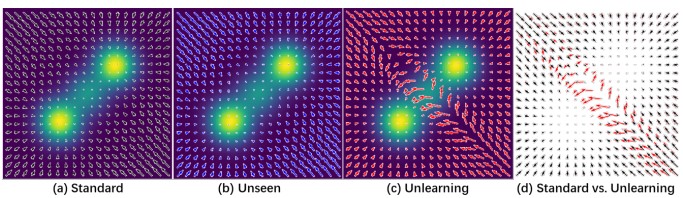

Figure 2: (a), (b) and (c) are the overlay of probability density of training data from $p_{\text{data}} = \frac{2}{5}\mathcal{N}((-2,-2),I) + \frac{1}{5}\mathcal{N}((0,0),I)\frac{2}{5}\mathcal{N}((2,2),I)$ and $s_{\theta}^{\mathbf{u}}(\mathbf{x}(0.08), 0.08)$, (d) comparison of the scores from our proposed method Unlearning and the Standard.

We conduct a quick experiment on the mixture Gaussian distribution using *Unlearning Re-training* strategy to evaluate the effectiveness of the proposed method. As shown in Figure 1, compared to *Unseen Re-training*, samples generated by our method almost do not contain NSFG data. Meanwhile, the NLL values in Table 1 indicate a substantial decrease in the probability of generating NSFG data. We further show why *Unlearning Re-training* can surpass *Unseen Re-training*. We plot the learned scores at a randomly selected generation process $t = 0.08$ in Figure 5. The results show that the scores for both *Unseen Re-training* and *Standard VESDE* are quite similar, while our method alters the score distribution of NSFG data, causing the model to steer away from high probability density areas, thereby reducing the likelihood of generating NSFG data.

## 4 EXPERIMENTS

### 4.1 EXPERIMENTAL SETUP

**Datasets Preparation.** We evaluate our proposed SGU on MNIST (Alsaafin & Elnagar, 2017), CIFAR-10 (Krizhevsky et al., 2009), STL-10 (Coates et al., 2011) and CelebA (Liu et al., 2015) datasets. Despite evaluation on score-based models such as Variance Preserving (VP) SDE (Song et al., 2021) and VE SDE (Song et al., 2021), we also employ DDPM (Ho et al., 2020) to verify the generalization of SGU to different types of diffusion generative models. According to the characteristics of the datasets, we conducted class forgetting experiments using MNIST (Alsaafin & Elnagar, 2017), CIFAR10 (Krizhevsky et al., 2009) and STL-10 (Coates et al., 2011) datasets, and performed attribute elimination generation on CelebA (Liu et al., 2015) datasets. We outline the dataset preparation for the experiments, detailing the selection of $\mathcal{D}_f$ and the generative models trained on each dataset as follows:

- MNIST: We trained the VE SDE model, selecting all instances of the digits "3" and "7" for $\mathcal{D}_f$. The MNIST dataset exhibits a sparse distribution of digits '0 − 9' in high-dimensional space. While individual handwritten digits are independent, they exhibit strong local structural dependencies. This characteristic makes the Variational SDE (VESDE) particularly suitable for modeling the MNIST data distribution.

- CIFAR-10: We trained the VP SDE and DDPM models, selecting the data labeled as "dog" and "automobile" classes for $\mathcal{D}_f$.

- STL-10: We trained the VP SDE models, selecting the data labeled as the "airplane" class for $\mathcal{D}_f$.

- CelebA:We trained the VP SDE model, selecting the feature "Bangs" from the 40 available features provided for each image to form $\mathcal{D}_f$.

**Compared Methods.** Our proposed `SGU` has two variants: `SGU-Orthogonal` and `SGU-Obtuse`. We compare `SGU`s with the following methods. *Standard*: the original generative models trained on $\mathcal{D}$ before unlearning serve as a reference. *Unseen*: a model retrained from scratch on data that does not contain $\mathcal{D}_f$. *Unseen* is often considered the 'gold standard' in MU and the state-of-the-art unlearning strategy.

**Evaluation Metric.** We use the following evaluation metrics to evaluate the effectiveness of unlearning:
Unlearning ratio(UR): UR measures the percentage of generated samples in the NSFG data produced by the model. A lower UR value indicates that the model has successfully unlearning the NSFG data. We use external classifiers or CLIP to evaluate the generated samples to ensure that the unlearning categories or attributes have been effectively removed. For all experiments, we randomly sample 10,000 images from the model to calculate the unlearning ratio.
Negative log-likelihood(NLL): For the SDE-based generation diffusion model, we can accurately calculate the value of NLL, from which we can calculate the likelihood of the generation of NSFG data and SFG data. Higher values indicate a lower probability of generation.

**Generation Quality Evaluation.** `SGU` preserves the generative quality when generating SFG data while generating noise to replace generating NSFG data. Therefore, using commonly used metrics FID (Heusel et al., 2017) and IS (Salimans et al., 2016) to evaluate generation performance is unsuitable, because these quality evaluation metrics assess the quality of whole generated data (including generated NSFG data and SFG data). We argue that we should evaluate the generative quality of generated NSFG and SFG data respectively. To this end, we test our method on image inpainting and reconstruction tasks, using CLIP embedding distance (Radford et al., 2021) to assess whether the reconstruction quality degrades on NSFG and SFG data.

### 4.2 CLASS-WISE/FEATURE-WISE UNGENERATION

**Quantitative Results.** In Table 2, we compare the unlearning performance with baseline methods in unconditional generation. First, `SGU` achieves the lowest unlearning rate compared to *Unseen* across all datasets, indicating that `SGU` effectively unlearns the NSFG data. Second, for *Unseen*, both SFG and NSFG data exhibit low NLL values, suggesting that despite the NSFG data never being observed during the training process, the generative model can still fit the distributions well. In contrast, `SGU` significantly reduces the generation probability of $D_f$ via substantially increasing the NLL values of the NSFG data. Additionally, although both `SGU-Orthogonal` and `SGU-Obtuse` can successfully unlearn undesirable data/features, their performance varies across different scenarios. `SGU-Orthogonal` is more effective for class unlearning, while `SGU-Obtuse` is more effective for feature unlearning. We suspect that `SGU-Orthogonal` seeks null space of $\nabla_{\mathbf{x}^f} \log p_t(\mathbf{x}^f)$, so that $s_\theta^{\mathbf{u}}(\mathbf{x}, t)$ does not learn any semantic features(see Figure 3), hence `SGU-Orthogonal` is effective for most cases. However, when $p_g(\mathbf{x})$ and $p_f(\mathbf{x})$ are very close, the null space of $\nabla_{\mathbf{x}^f} \log p_t(\mathbf{x}^f)$ is hard to be found, hence using `SGU-Orthogonal` to extend the search space ($s_\theta^{\mathbf{u}}(\mathbf{x}^f(t), t) \cdot \nabla_{\mathbf{x}^f} \log p_t(\mathbf{x}^f) < 0$, $\mathbf{x}^f \in \mathcal{D}_f$) can improve the unlearning performance.

**Qualitative Results.** We report the qualitative visualization comparison in Figure 3. In Figure 3, we observe that *Unseen* may not completely erase the bangs features. For example, facial images generated by *Unseen* may still exhibit few bangs features, even though the bang features are not as long as those in $D_f$. In contrast, `SGU` completely erases the bang features. An interesting phenomenon is that `SGU-Orthogonal` and `SGU-Obtuse` forget bangs in different ways. For the unwanted feature, `SGU-Orthogonal` replaced the bangs with noisy images, while `SGU-Obtuse` generate features opposite to the bangs in the score distribution, such as 'no bangs' or 'hat'. This is because `SGU-Orthogonal` seeks for null space of $\nabla_{\mathbf{x}^f} \log p_t(\mathbf{x}^f)$, hence $s_\theta^{\mathbf{u}}(\mathbf{x}, t)$ learns nothing, while $s_\theta^{\mathbf{u}}(\mathbf{x}, t)$ in `SGU-Obtuse` learn the inverse of $\nabla_{\mathbf{x}^f} \log p_t(\mathbf{x}^f)$, hence may generate the 'inverse' feature of bangs. The visual results in other datasets also have the same phenomenon, as shown in

Table 2: Quantitative results for unleaning feature or class on different datasets. 'Feature/class' means we need to unlearn content. The unlearning ratio represents the degree of forgetting, measured by predicting the proportion of $D_f$ data in the generated 10,000 images using CLIP. The right side of the table presents the negative log-likelihood values for $D$, $D_g$ and $D_f$ data.

| Dataset | Model | Feature/class | Unlearning ratio (%) (↓) | | | | Test | Negative log-likelihood (NLL) Test (↓) | | | |
|---|---|---|---|---|---|---|---|---|---|---|---|
| | | | Standard | SGU-Orthogonal | SGU-Obtuse | Unseen | | Standard | SGU-Orthogonal | SGU-Obtuse | Unseen |
| MNIST | VESDE | 3 | 11.0 | **0.4** | 1.5 | 1.8 | $\mathcal{D}_g$ | 2.82 | 3.92 | 3.70 | 3.07 |
| | | 7 | 15.8 | **0.8** | 3.6 | 2.3 | $\mathcal{D}_f$ | 2.78 | 13.23 | 12.08 | 3.01 |
| | | 3 and 7 | 26.8 | **1.2** | 5.1 | 4.1 | | | | | |
| CIFAR-10 | VPSDE | automobile | 11.2 | 1.9 | **0.9** | 3.4 | $\mathcal{D}_g$ | 3.12 | 3.22 | 3.28 | 3.09 |
| | | dog | 13.4 | **10.0** | 11.5 | 10.8 | $\mathcal{D}_f$ | 3.20 | 5.94 | 4.37 | 3.21 |
| | | dog and automobile | 24.6 | **11.9** | 12.4 | 14.2 | | | | | |
| STL-10 | VPSDE | airplane | 12.1 | **2.4** | 3.6 | 3.8 | $\mathcal{D}_g$ | 2.90 | 2.90 | 2.92 | 2.90 |
| | | | | | | | $\mathcal{D}_f$ | 2.19 | 8.94 | 9.25 | 2.32 |
| CELEBA | VPSDE | Bangs | 19.6 | 3.5 | **0.7** | 6.7 | / | / | / | / | / |

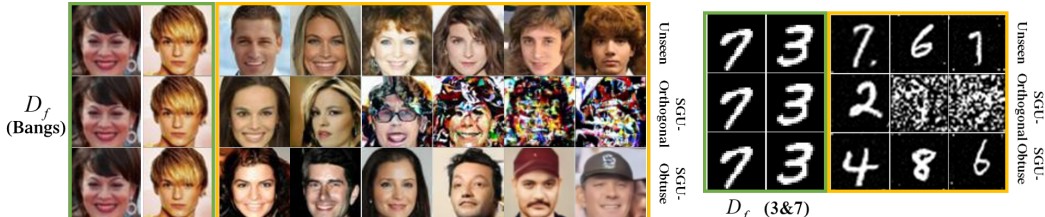

Figure 3: Image generation using different unlearning methods for VP SDE on MNIST and CELEBA. The top, middle, and bottom rows show images generated by unlearning strategy *Unseen*, `SGU-Orthogonal` and `SGU-Obtuse` respectively. NSFG images sampled from the forgetting dataset $D_f$ are enclosed in the green box. Images generated by the different unlearning methods are enclosed in the yellow box.

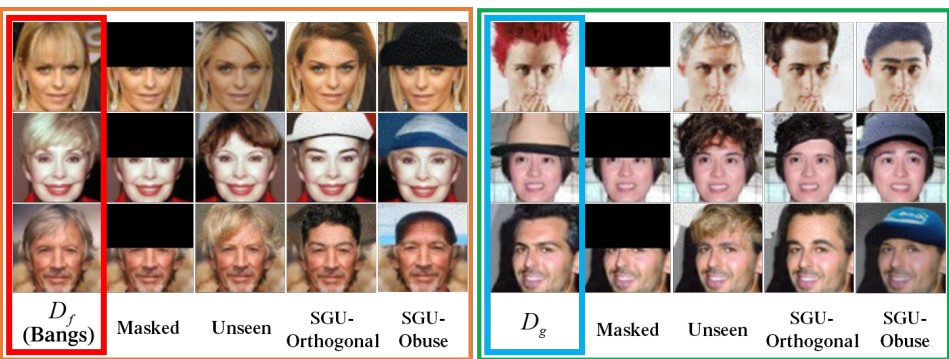

Figure 4: The comparison of restoration results on the CELEBA dataset. The mask size is $64 \times 32$, in the upper half of the image. The restored results on $D_f$ are displayed on the left, enclosed in the orange box. The restored results on $D_g$ are displayed on the right, enclosed in the green box.

the right side of the Figure 3. Additionally, for SFG content generation, `SGU` shows competitive generative performance compared to the source images, and performs well with high-resolution images.

### 4.3 APPLICATION TO DOWNSTREAM TASKS

**Unleanring Inpainting.** `SGU` enables zero-shot transfer of the unlearning SGM to downstream task. We first test `SGU` on inpainting task. We mask the upper part of the image and attempt to restore the whole image. The quantitative restoring results on $D_f$ and $D_g$ are reported in Section 4.2. We regard the classification as correct if the predicted class of the restored image matches that of the corresponding original image. `SGU-Obtuse` still contains a high classification accuracy for restored images on $D_g$ while significantly decrease the accuracy on restored images on $D_f$. This indicates that restored image by `SGU-Obtuse` still retains similar semantics on $D_g$, while altering

Table 3: The performance results of various methods on the inpainting task of Original, Orthogonal, Obtuse and Unseen. "Clean" refers to the prediction accuracy on real dataset using the CLIP classifier. 'Classification accuracy' means if the predicted class of the restored image matches that of the corresponding original image. 'CLIP of $D_g$' indicates the CLIP distance between the restored and original images for 5,000 images after the inpainting task.

| Dataset | Feature/class | | Classification accuracy (%) | | | | | CLIP of $D_g$($\downarrow$) | | | |
|---|---|---|---|---|---|---|---|---|---|---|---|
| | | | Clean | Standard | SGU-Orthogonal | SGU-Obtuse | Unseen | Standard | SGU-Orthogonal | SGU-Obtuse | Unseen |
| CIFAR-10 | dog and automobile | $D_g$ | 95.4 | 72.5 | 75.5 | 74.7 | 75.8 | 6.80 | 6.80 | 6.77 | 6.72 |
| | | $D_f$ | 95.5 | 75.0 | 57.2 | 49.6 | 59.7 | | | | |
| STL-10 | airplane | $D_g$ | 96.3 | 83.4 | 83.6 | 83.1 | 84.5 | 8.50 | 8.51 | 8.50 | 8.50 |
| | | $D_f$ | 96.3 | 84.1 | 59.5 | 50.3 | 54.9 | | | | |

Table 4: The comparison results of reconstruction on CIFAR-10.

| Dataset | Model | | Classification accuracy (%) | | | | | CLIP | | | |
|---|---|---|---|---|---|---|---|---|---|---|---|
| | | | Standard | SGU-Orthogonal | SGU-Obtuse | Unseen | | Standard | SGU-Orthogonal | SGU-Obtuse | Unseen |
| CIFAR-10 | VPSDE | $\mathcal{D}_g$ | 88.1 | 87.7 | 87.0 | 87.9 | $\mathcal{D}_g$ | 6.91 | 6.90 | 6.89 | 6.90 |
| | | $\mathcal{D}_f$ | 74.4 | 48.4 | 69.6 | 70.3 | $\mathcal{D}_f$ | 7.02 | 7.25 | 7.00 | 7.00 |

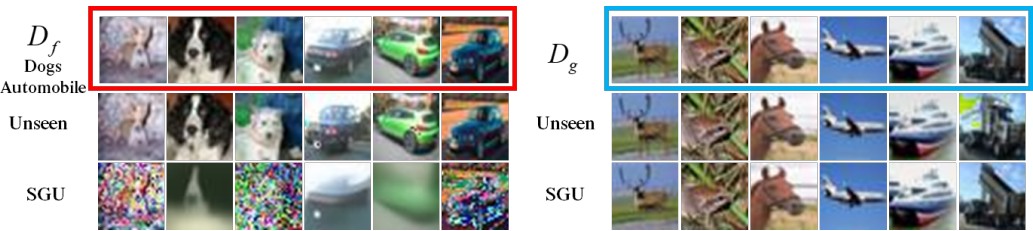

Figure 5: The comparison of reconstruction results on the CIFAR-10 dataset. The top, middle and bottom columns are the original images, reconstruction images by *Unseen*, and reconstruction images by SGU-Orthogonal respectively.

the source semantics on $D_g$. In addition, SGU captures the same CLIP distance as a standard trained generator on $D_g$, indicating that the SGU-trained generator still retains high generation performance. Furthermore, we compare the visual results on Figure 4. When the masked image is from $D_g$ (no 'bangs'), *Unseen* still has the probability to restore a face image with bangs. When the masked image is from $D_f$, most of the restored image fail to erase the 'bang' features. In contrast, our method can effectively erase the bangs on $D_f$, and restore the similar semantic features on $D_g$

**Unlearning Reconstruction.** Generative models can learn the latent representations of data and reconstruct images. Through the reconstruction, we use these latent representations as guidance to verify whether our method effectively achieves unlearning. To maintain the similarity between reconstruction results and original images on $D_g$, we set $t = 0.02$ for the continuous-time SDE schedule. We reconstruct images using VP SDE model trained by standard training, *Unseen* and our proposed SGU method and report the comparison results in Table 4. We utilize the classification accuracy to assess whether the reconstructed images still be classed by the original class. SGU-Obtuse significantly decrease the accuracy for reconstructed $D_f$ data while maintain the original semantic information for reconstructed $D_g$. Additionally, we calculate the CLIP distance for $D_g$ and $D_f$ with respect to their respective ground truth images. Our method SGU-Orthogonal demonstrates superior forgetting effects compared to the *Unseen*, with a larger CLIP distance on $D_f$. Next, we visualize the reconstruction quality in Figure 5. Unlike *Unseen*, where the reconstruction quality of $D_f$ matches that of $D_g$, SGU-Orthogonal reconstructs $D_f$ as noisy images, indicating that SGU-Orthogonal has completely unlearned the $D_f$ distribution.

## 4.4 UNLEARNING DDPM AND FINE-TUNE

SGU is a general and flexible framework that is compatible with DDPM models and fine-tuning training. The technical details of SGU application to DDPM can be found in Appendix B. To demonstrate this, we conduct both class and feature unlearning on pre-trained VP SDE and DDPM models. The Table 5 presents quantitative results for fine-tuning experiments on different datasets using the SGU method. We conduct 80,000 and 30,000 iterations of fine-tuning on SGM and DDPM architecture respectively, across all datasets. It is noteworthy that SGU also performed well in the

DDPM fine-tuning tasks, indicating that our unlearning framework can also be applied to the DDPM architecture. `SGU-Obtuse` achieve the lowest unlearning rates across all models and datasets, indicating its effectiveness for erasing undesirable content from pre-trained models. Since the DDPM architecture cannot calculate exact NLL values, they are marked as '\' in the table. Similarly, on the CelebA dataset, where the unlearning task involves removing attribute features, NLL values on image data cannot reflect the probability of generating specific attributes, and thus are also marked as '\'. `SGU` surpasses *Unseen* in NLL tests.

Table 5: Fine-tune quantitative results for unleaning feature or class on different datasets. 'Feature/class' means we need to unlearn content. The unlearning ratio represents the degree of forgetting, measured by predicting the proportion of $D_f$ data in the generated 10,000 images using CLIP. The right side of the table presents the negative log-likelihood values for $D$, $D_g$ and $D_f$ data.

| Dataset | Model | Feature/class | Unlearning ratio (%) (↓) | | | | Test | Negative log-likelihood Test (↓) | | | |
|---|---|---|---|---|---|---|---|---|---|---|---|
| | | | Standard | SGU-Orthogonal | SGU-Obtuse | Unseen | | Standard | SGU-Orthogonal | SGU-Obtuse | Unseen |
| CIFAR-10 | VPSDE | automobile | 11.2 | 2.7 | **0.6** | 3.4 | $\mathcal{D}_g$ | 2.89 | 3.06 | 4.36 | 2.92 |
| | | dog | 13.4 | **8.7** | 8.9 | 10.8 | | | | | |
| | | dog and automobile | 24.6 | 11.4 | **9.5** | 14.2 | $\mathcal{D}_f$ | 2.91 | 10.36 | 14.96 | 2.95 |
| | DDPM | automobile | 13.1 | 3.3 | **1.6** | 2.7 | \ | \ | \ | \ | \ |
| | | dog | 13.9 | 5.4 | **3.6** | 4.5 | \ | \ | \ | \ | \ |
| | | dog and automobile | 27.0 | 8.7 | **5.2** | 7.2 | \ | \ | \ | \ | \ |
| CELEBA | VPSDE | Bangs | 19.6 | 2.6 | **0.1** | 6.7 | \ | \ | \ | \ | \ |

## 5 CONCLUSION

In this work, we make the first attempt to investigate generative unlearning in score-based generative model. To this end, we introduce score-based generative unlearning (`SGU`), which surpass the previous 'gold standard' for machine unlearning in score-based generative models. Extensive experiments demonstrate that `SGU` effectively unlearns undesirable content, without sacrificing generation quality for suitable data. Although SGC is primarily designed for score-based generative model, `SGU` is a straightforward and flexible unlearning framework, which can be generalized to diverse diffusion architectures (SGM and DDPM) and training strategies (re-training and fine-tune). Additionally, `SGU` effectively enables zero-shot transfer of the unlearning score-based generative model to downstream tasks, including image inpainting and reconstruction. This further illustrates that `SGU` maintains effective unlearning even when faced with inappropriate content guidance.

**Ethics Statement.** The application of Score-based Generative Unlearning (`SGU`) for unconditional generative models can be effectively utilized to prevent the generation of content related to user privacy and copyright violations. Moreover, `SGU` can mitigate the risk of producing harmful content, such as violence or pornography, ensuring a more responsible and ethical use of generative technologies.

**Reproducibility Statement.** For the datasets used in our experiments, all the datasets used in this paper are open dataset and are available to the public. Besides, our codes are primarily based on Pytorch. All the source code and model checkpoints will be shared upon acceptance. All inference details and mathematical deduction can be found in Section 3.

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

## A    RELATED WORK

**Score-based generative models**    In recent years, there are two classes of diffusion generative models that sequentially corrupt training data with slowly increasing noise and then learn to reverse this corruption. *Score matching with Langevin dynamics* (SMLD)(Song & Ermon, 2019; 2020) estimates the *score* at each noise scale and then employs Langevin dynamics to sample from a sequence of progressively reduced noise scales during the generation process. *Denoising diffusion probabilistic modeling* (DDPM) (Ho et al., 2020; Nichol & Dhariwal, 2021; Sohl-Dickstein et al., 2015) is a probabilistic generative model that learns the distribution of original data by incrementally adding noise to the data to degrade the data structure, and then learning a corresponding reverse process to denoise it. Since the training objective of DDPM implicitly computes the *score* at each noise scale in a continuous state space, we can refer to these two types of models as *score-based generative models* or *diffusion models*. To develop new sampling methods and further enhance the capabilities of score-based generative models, a unified framework is introduced that generalizes previous approaches through the lens of stochastic differential equations (SDEs) (Song et al., 2021). Specifically, this framework considers perturbing the data based on a continuous distribution evolving over time according to a diffusion process, rather than relying on a finite number of noise distributions.

Due to the powerful performance of the diffusion model, many diffusion models are used to generate creative content in actual production. Applications for image generation systems like DALL · E2(Ramesh et al., 2022), Imagen(Saharia et al., 2022) and Stable Diffusion(Rombach et al., 2022a) utilize diffusion models for conditional generation. More recently, some applications have started to employ diffusion models to generate various modal data such as voice and video. But at the same time, the powerful capabilities and wide application of the diffusion model have also led to potential negative impacts. The content generated by the diffusion model has copyright issues, privacy issues, bias issues, etc.

**Security and privacy of AIGC**   The emergence of AIGC marks a critical moment in the evolution of information technology. With AIGC, generating high-quality data has become easy. However, the surge in generated data in cyberspace also brings security and privacy issues, including personal privacy leakage and media forgery, which may be used for improper purposes such as fraud. Diffusion models, as a major type of deep generative model, typically use training data from various open sources. When utilizing such unfiltered data, there is a risk of contamination (Chen et al., 2023b) or manipulation (Rando et al., 2022), which may lead to the generation of inappropriate content (Schramowski et al., 2023). Additionally, these models may risk imitating copyrighted content, such as replicating artistic styles (Gandikota et al., 2023; Shan et al., 2023).

**Machine unlearning**   MU can help generative models forget training data. To address the challenges in the security and privacy of AIGC, it is urgent to explore effective MU techniques. However, SALUN (Fan et al., 2024) demonstrated that existing MU methods designed for image classification are not sufficient to address MU in image generation. Therefore, the challenge of ensuring effective unlearning for generative models has become increasingly important and pressing. Recently, a few studies (Gandikota et al., 2023; Heng & Soh, 2023; Fan et al., 2024; Zhang et al., 2024; Kumari et al., 2023; Wu et al., 2024; Heng & Soh, 2024) have explored unlearning in diffusion models, with most focusing on text-to-image diffusion models. Kumari et. al (Kumari et al., 2023) fine-tuned diffusion models by modifying the sensitive training data so that the models forget already memorized images. Forget-Me-Not (Zhang et al., 2024) is adapted as a lightweight model patch for Stable Diffusion. It effectively removes the concept of containing a specific identity and avoids generating any face photo with the identity. SA (Heng & Soh, 2024) can be applied to conditional variational likelihood models, which encompass a variety of popular deep generative frameworks, including variational autoencoders and large-scale text-to-image diffusion models. ERASEDIFF (Wu et al., 2024) explores the issue of unlearning in diffusion models and proposes an effective unlearning method for both unconditional and conditional diffusion models. While current research provides strategies for concept erasure in diffusion models, achieving precision comparable to exact forgetting remains a challenging task.

## B   SGU FOR UNLEARNING DDPM

Denoise Diffusion Probabilistic models (DDPMs) (Ho et al., 2020) are a type of generative model that generate samples from a distribution via an iterative Markov denoising method. Initially, a sample $x_T$ is drawn from a Gaussian distribution and subsequently denoised over $T$ time steps, ultimately resulting in a clean sample $x_0$. During the training phase, the model learns to predict the noise $\epsilon_\theta(x_t, t)$ that needs to be removed from the sample $x_t$ using the following reweighted variational bound:

$$L_{USGM(g)} = \mathbb{E}_{\mathbf{x}_0, \epsilon}\left[\frac{\beta_t^2}{2\sigma_t^2 \alpha_t(1 - \bar{\alpha}_t)} \left\|\epsilon - \epsilon_\theta(\sqrt{\bar{\alpha}_t}\mathbf{x}_0^g + \sqrt{1 - \bar{\alpha}_t}\epsilon, t)\right\|^2\right], \ \mathbf{x}_0^g \in \mathcal{D}_g, \qquad (12)$$

where $\beta_1, \ldots, \beta_T$ is a variance schedule used for adds Gaussian noise to the data in the forward process, $\alpha_t = 1 - \beta_t$, $\bar{\alpha}_t = \prod_{s=1}^{t} \alpha_s$ and $x_t = \sqrt{\bar{\alpha}_t}x_0 + \sqrt{1 - \bar{\alpha}_t}\epsilon$ for $\epsilon \sim \mathcal{N}(0, I)$. While our method is primarily designed for score-based generative models, the Score-based Generative Unlearning (SGU) approach is also compatible with the DDPM models. By applying our method within the DDPM framework, we derive the following unlearning method:

**SGU-Orthogonal for DDPM**

$$L_{USGM(f)} = \mathbb{E}_{\mathbf{x}_0, \epsilon}\left[\frac{\beta_t^2}{2\sigma_t^2 \alpha_t(1 - \bar{\alpha}_t)} \left\|\epsilon \cdot \epsilon_\theta^{\mathbf{u}}(\sqrt{\bar{\alpha}_t}\mathbf{x}_0^f + \sqrt{1 - \bar{\alpha}_t}\epsilon, t)\right\|^2\right], \ \mathbf{x}_0^f \in \mathcal{D}_f. \qquad (13)$$

**SGU-Obtuse for DDPM**

$$L_{USGM(f)} = \mathbb{E}_{\mathbf{x}_0, \boldsymbol{\epsilon}} \left[ \frac{\beta_t^2}{2\sigma_t^2 \alpha_t (1 - \bar{\alpha}_t)} \left( \boldsymbol{\epsilon} \cdot \boldsymbol{\epsilon}_\theta^{\mathbf{u}} (\sqrt{\bar{\alpha}_t} \mathbf{x}_0^f + \sqrt{1 - \bar{\alpha}_t} \boldsymbol{\epsilon}, t) \right) \right], \ \mathbf{x}_0^f \in \mathcal{D}_f. \quad (14)$$

Similarly, the final loss of unlearning DDPM modeling can be solved by Equation (11).

