# OpenReview forum: "UNLEARNING IS BETTER THAN UNSEEN: UNLEARNING SCORE-BASED GENERATIVE MODEL"
_ICLR.cc/2025/Conference — ICLR 2025 Conference Withdrawn Submission_

### Official Review · Reviewer_xmaD · 2024-10-31

**Soundness:** 3
**Presentation:** 4
**Contribution:** 2
**Rating:** 3
**Confidence:** 3

**Summary:**

This paper proposes the first Score-based Generative Unlearning (SGU) method for SGM to address privacy concerns and the potential misuse of AI-generated content, thereby surpassing the previous 'gold standard' of unlearning.

**Strengths:**

1. The motivation for this paper is clear.
2. The authors provide a detailed analysis of both the motivation and their proposed score-based generative unlearning method.
3. The writing is well-structured.

**Weaknesses:**

1. The novelty is limited. The authors only design a score-based generative unlearning method to improve generation quality.
2. Given the existence of many advanced generative models, the authors focus solely on the basic DDPM and some score-based models. This raises the question of whether using more data and better models (such as Stable Diffusion XL) could solve the issues related to desirable generation content, making the authors' proposed method unnecessary.
3. Nowadays, there are also many autoregressive generation models. Is the proposed score-based generative unlearning method possibly extensible to other architectures?

**Questions:**

Please refer to the Weakness section.

---

### Official Review · Reviewer_Taty · 2024-10-31

**Soundness:** 2
**Presentation:** 3
**Contribution:** 1
**Rating:** 3
**Confidence:** 3

**Summary:**

The authors propose to use an unlearning method for score-based generative models. Prior work on this research topic normally focus on DDPM. They argue that the unlearning method is better than the gold standard in this area -- retraining a model from scratch after removing the undesirable data. The algorithm is tested on standard datasets and leads to decent performance.

**Strengths:**

1. The research topic of privacy and security of generative models is relevant and timely.
2. The paper is well-written and easy to follow.
3. The experiments are extensive.

**Weaknesses:**

1. I don’t understand why the unlearning method for SGM should be different from DDPM. Actually, it is not necessary to differentiate these two models, they essentially learn a score function by deep neural networks. As [1] argues that DDPM could be seen as VP-SDE.
2. The arguments about the conditional unlearning and unconditional unlearning is confusing, actually a text-to-image model could also generate unconditional images, as long as the prompt is null.
3. As shown in Fig 4, it seems that the proposed unlearning method does not significantly outperform the unseen method.


[1] SCORE-BASED GENERATIVE MODELING THROUGH STOCHASTIC DIFFERENTIAL EQUATIONS. ICLR2021

**Questions:**

See weakness. I am willing to raise the score if my concerns can be well-addressed.

---

### Official Review · Reviewer_hrCq · 2024-11-03

**Soundness:** 2
**Presentation:** 2
**Contribution:** 2
**Rating:** 3
**Confidence:** 4

**Summary:**

This paper highlights the need for effective unlearning techniques in diffusion models. Traditional methods, which rely on retraining models from scratch, tend to perform poorly in score-based generative models. To address this limitation, the authors propose Score-based Generative Unlearning (SGU), which introduces a novel score adjustment strategy during the continuous-time stochastic process to diverge from undesirable data scores. Experimental results demonstrate that SGU significantly reduces unwanted content generation while preserving high-quality outputs and enabling zero-shot transfer across various tasks.

**Strengths:**

A key strength of this paper is its novel approach in proposing "Unlearning Re-training" instead of the traditional "Unseen Re-training" method. This innovative methodology offers a fresh perspective on efficiently approximating the target score distribution for unlearning purposes. The authors provide valuable intuition through synthetic experiments, illustrating why this novel algorithm better aligns with the desired score distribution, thereby significantly improving performance in unlearning unwanted content.

**Weaknesses:**

While the proposed approach shows promise, several aspects raise questions about its robustness and effectiveness:

1. Preservation of Generation Quality : There is some uncertainty about whether the model can reliably generate images without the undesirable features while maintaining overall generation quality. Although the paper presents classification accuracy or CLIP classifier results on inpainting and reconstruction tasks to demonstrate the preservation of the original model's performance, these evaluations may be insufficient. Before discussing downstream task performance, it would be beneficial to assess the generation quality of the modified model more thoroughly. For instance, calculating the FID score between the generated data and training data (excluding NSFG data) could provide insights into how well the model retains quality.

2. Scalability to Large-Scale, High-Resolution Data: The applicability of the proposed method to large-scale, high-resolution datasets remains unclear. To better understand the approach's effectiveness, additional experiments on datasets with a greater number of classes, such as ImageNet, could demonstrate whether the method can maintain performance and reliability at scale.

**Questions:**

While the paper provides a comprehensive overview of the proposed approach, there are some areas that would benefit from additional clarification:

1. Determination of the Alpha Value in Equation (11) : Further explanation is needed regarding how the value of alpha in Equation (11) was chosen. The expression "depends on the ratio of M to N" is somewhat unclear, and it would be helpful to understand what values M and N correspond to. If alpha represents the proportion of D_f within the overall dataset, it would be beneficial to provide the specific alpha values used for each experiment in the results section to enhance clarity.

2. Effectiveness of the Algorithm Relative to D_f Proportion : It would be interesting to understand up to what percentage of the dataset D_f
can occupy before the algorithm's effectiveness is compromised. Specifically, if D_f constitutes a large portion of the data, can the model still achieve stable training by reducing its influence through adjustments to alpha? Clarification on this point would help in evaluating the algorithm's robustness under varying data distributions.

---

### Official Review · Reviewer_cdHR · 2024-11-04

**Soundness:** 3
**Presentation:** 3
**Contribution:** 3
**Rating:** 8
**Confidence:** 3

**Summary:**

This paper presents a novel Score-based Generative Unlearning (SGU) framework aimed at enhancing the unlearning capabilities of Score-Based Generative Models (SGM) by introducing methods that outperform the "gold standard" retraining approach in mitigating the generation of undesirable data. SGU implements a score adjustment process to modify the learned scores of the unwanted data during training, resulting in minimal generation of undesirable content while maintaining high-quality generation for desirable data. The proposed approach, compatible with both SGM and DDPM models, supports transfer to downstream applications like image inpainting and reconstruction, demonstrating effective unlearning across multiple datasets.

**Strengths:**

1. The paper addresses a critical gap in machine unlearning research for Score-Based Generative Models, offering a more effective alternative to standard retraining approaches.
2. SGU showcases strong flexibility, being adaptable to multiple generative model architectures (e.g., SGM and DDPM) and training strategies.
3. The method demonstrates consistent and high performance in removing undesirable content across a range of datasets and tasks, including zero-shot transfer to image inpainting and reconstruction.

**Weaknesses:**

1. The method may face optimization challenges, especially when the undesirable and desirable data distributions are similar, which might complicate the unlearning process.
2. The paper offers limited discussion on potential trade-offs between unlearning effectiveness and computational efficiency, such as the additional overhead introduced by using SGU compared to its baselines.
3. The study focuses on class-level unseen data, where defining boundaries for unlearning is relatively straightforward. Discussing potential challenges or limitations in applying the method to more granular or ambiguous unlearning tasks beyond class-level data would provide valuable insights into the method's broader applicability and limitations.

**Questions:**

Please see the weaknesses.

---

### Note · Authors · 2024-11-12

I have read and agree with the venue's withdrawal policy on behalf of myself and my co-authors.